# The Diagnostic, Therapeutic and Prognostic Relevance of Neutrophil Extracellular Traps in Polytrauma

**DOI:** 10.3390/biom13111625

**Published:** 2023-11-07

**Authors:** Emily Rogers, Shevani Pothugunta, Veronika Kosmider, Natasha Stokes, Layla Bonomini, Gabrielle D. Briggs, Daniel P. Lewis, Zsolt J. Balogh

**Affiliations:** 1School of Medicine and Public Health, University of Newcastle, Newcastle, NSW 2308, Australiashevani.pothugunta@uon.edu.au (S.P.);; 2Discipline of Surgery, School of Medicine and Public Health, University of Newcastle, Newcastle, NSW 2308, Australia; gabrielle.briggs@newcastle.edu.au (G.D.B.);; 3Injury and Trauma Research Program, Hunter Medical Research Institute, Newcastle, NSW 2308, Australia; 4Department of Traumatology, John Hunter Hospital, Newcastle, NSW 2308, Australia

**Keywords:** trauma, polytrauma, neutrophil extracellular traps, NET, traumatic shock, trauma centre, mitochondrial DNA, Damage associated molecular patterns, DAMPs

## Abstract

Neutrophil extracellular traps (NETs) represent a recently discovered polymorphonuclear leukocyte-associated ancient defence mechanism, and they have also been identified as part of polytrauma patients’ sterile inflammatory response. This systematic review aimed to determine the clinical significance of NETs in polytrauma, focusing on potential prognostic, diagnostic and therapeutic relevance. The methodology covered all major databases and all study types, but was restricted to polytraumatised humans. Fourteen studies met the inclusion criteria, reporting on 1967 patients. Ten samples were taken from plasma and four from whole blood. There was no standardisation of methodology of NET detection among plasma studies; however, of all the papers that included a healthy control NET, proxies were increased. Polytrauma patients were consistently reported to have higher concentrations of NET markers in peripheral blood than those in healthy controls, but their diagnostic, therapeutic and prognostic utility is equivocal due to the diverse study population and methodology. After 20 years since the discovery of NETs, their natural history and potential clinical utility in polytrauma is undetermined, requiring further standardisation and research.

## 1. Background

Injury is a leading cause of disability and mortality globally, resulting in approximately 5 million deaths per year [1]. Trauma leads to an inflammatory response through either direct tissue injury or tissue hypoxia. An acute inflammatory response (8–10 days) can be self-limiting and non-specific, occurring almost immediately after traumatic insult; however, when this response is not contained, this can lead to adverse events. Those who survive traumatic events are at high risk of developing systemic inflammatory response syndrome (SIRS) and multiple organ Failure (MOF), as well as other complications such as hypercoagulability and deep vein thrombosis (DVT). SIRS and MOF are characterised clinically by elevated concentrations of inflammatory mediators triggered by damaged/necrotic tissue, dysregulated inflammatory cell function, hypermetabolism and maldistributions of circulation volume [2]. Many similarities exist between the mechanisms and downstream manifestations of post-injury inflammation and systemic inflammation in sepsis, including Toll-like receptor signalling, cytokine storms, neutrophil and platelet activation, dysfunctional coagulation and multiple organ failure as global homeostatic dysfunction. MOF in trauma patients can also develop in response to a septic complication, classically described as late MOF. As early trauma care improves and more critically injured patients survive, the importance of understanding and managing the post-injury inflammatory response is critical [3].

Neutrophils play a key role in inflammation and cellular defence, accounting for 50–70% of circulating leukocytes. They are essential to the innate immune response. Normally, neutrophils prevent tissue damage through defence mechanisms targeted against microorganism invasion such as margination, diapedesis, phagocytosis and chemotaxis [2,3]. Other defence mechanisms include cellular signalling to various receptors involved in the recognition of and response to injury or infection, antimicrobial molecule production, the prevention of infectious spread and the homeostatic directive role in immune response (both innate and adaptive). Additionally, neutrophils can be associated with collateral tissue damage, delayed healing and dysfunctional inflammatory response [1,3].

In the event of severe injury, neutrophils are activated systemically and can cause significant damage to previously uninjured tissues. This occurs via systemic interstitial neutrophil infiltration and accumulation, resulting in the local release of cytotoxic and histologic enzymes, including myeloperoxidase (MPO) and elastase. This dysregulated response contributes to the systemic inflammatory response syndrome (SIRS) characterised by microvascular damage, oedema and microthrombosis, and frequently leads to post-injury multiple organ failure (MOF) [4].

Neutrophil extracellular traps (NETs) are one of the first-line defence mechanisms of the innate immune system, first described in 2004 [4]. Their formation occurs through the process of NETosis, a type of regulated cell death distinguishable from apoptosis and necrosis. They are complex systems composed of extruded nuclear or mitochondrial DNA in a scaffold of proteases and various inflammatory mediators [5] forming a web-like structure. The role of NETs in the innate immune system is to trap and immobilise cells and pathogens as part of a wider immune response and phagocytosis. Initially, NETs were thought to be released exclusively in response to pathogenic microorganisms; however, the recent literature has shown the presence of NETs in trauma without infection [6]. Additionally, they have also been found to be present in other non-infectious acute and chronic inflammatory conditions, such as breast and gastrointestinal malignancies, acute respiratory distress syndrome, systemic lupus erythematous and deep vein thrombosis [5,7,8].

The quantification of NETs in clinical samples typically involves measuring their unique components in plasma or in neutrophils that have undergone NETosis ex vivo, including cell-free DNA (cf-DNA), neutrophil-derived proteins known to exist in NETs, such as MPO and neutrophil elastase, and particularly citrullinated histone 3 (citH3), which is a modification of histone 3 that occurs specifically during the unwinding of DNA necessary for NETosis.

Predictive biomarkers to identify at-risk patients for post-injury inflammatory complications or to time therapeutic interventions can optimise patient management. Furthermore, they have the potential to act as an important therapeutic target in post-injury complications associated with hyperinflammation, such as sepsis and MOF. NETs are a potentially promising target for prediction, prognostication and therapeutic interventions for Polytrauma patients.

## 2. Aims

The aim was to investigate the diagnostic, therapeutic and prognostic relevance of NETs in severe trauma.

## 3. Methods

Preferred Reporting Items for Systematic Reviews and Meta-Analyses (PRISMA) was followed, and the methods were registered with PROSPERO, an international prospective register of systematic reviews; PROSPERO ID: CRD42023393012.

### 3.1. Search Strategy

Studies were identified through an extensive search of the PUBMED, EMBASE and COCHRANE databases (2004 to 14 April 2023). The search strategy included a mix of MeSH and free text terms for the key concepts relevant to neutrophil extracellular traps and trauma. The MeSH terms were (Neutrophil extracellular traps or extracellular traps or NETs or NETosis) and (Injury or trauma). Due to the paucity of this research topic, the search strategies were intentionally broad to increase the number of relevant studies identified. 

Only manuscripts published after the discovery of NETs in 2004 were included in the search; additionally, only manuscripts in English or with available English translations were eligible. Reference lists of key included papers and the latest editions of relevant journals were reviewed for new references. Full articles were read and assessed by two reviewers for relevance and study eligibility using Covidence. Disagreements over methodology were resolved via discussion, and a third reviewer, ZJB, adjudicated over any dispute. 

### 3.2. Study Selection

The included manuscripts examined the presence of NETs in severely traumatised patients (injury severity score > 15), the association between NETs and prognosis, and potential treatments/interventions for NETs in trauma patients. No restriction was placed on the method of NET detection or quantification, or the type of surgical or medical intervention the patient received. Only data relating to trauma patients were included. Due to the novelty and exploratory nature of the topic, all study types were eligible. After initial screening, duplicate data sets and articles such as editorials and discussion papers that did not match the inclusion criteria were excluded. 

### 3.3. Outcome Measures

Due to the relative scarcity of studies on this topic, the primary aim was to summate the literature relating to NETs in trauma. The primary outcome of interest was the relationship between NETs and poor prognosis, mortality and the incidence of MOF. The goal was to highlight potential interventions for NETs in trauma patients and to demonstrate the relationship between NETs and thrombus formation in traumatised patients. Data relating to study design, country of origin, patient characteristics, mechanism of injury, ISS, methodology of NET quantification and diagnosis, any therapies used in the studies and clinical outcome data such as those on the mortality of adverse outcomes were extracted.

### 3.4. Assessment of Study Quality

Quality assessment was performed using the validated tool designed by Guo et al., 2016, a 20-component checklist specifically designed for the quality assessment of case-control and retrospective studies. Guo et al.’s 2016 scoring system does not assign a minimum score in which a certain level of quality is achieved, but instead each of the criteria is weighed equally [9].

## 4. Results

Two-thousand one-hundred and forty manuscripts were identified through the search strategy (Figure 1). From this, 364 duplicates were removed before 1776 abstracts were screened, and 1711 were excluded as they were not relevant to our aim. Fourteen studies were included with seventeen excluded based on patient populations, thirteen were excluded based on non-trauma populations, fifteen were excluded based on study design, five were excluded based on outcomes, and one was excluded as only the abstract was accessible. These 14 studies all reported on NETs following severe traumatic injury, with 10 measuring them from plasma and four measuring ex vivo NETs. Eleven studies were prospective cohort studies, one was a retrospective cohort study, one was a clinical trial, and one was a post hoc analysis. Using the tool by Guo et al. for quality assessment, one study scored 12 [10], four studies scored 13 [8,11,12,13], four studies scored 14 [14,15,16,17], two studies scored 15 [18,19], two studies scored 16 [20,21], and one study scored 17 [22].

In total, the pooled sample was 1967 patients. The mean age was 44.12, where reported. Ten studies reported male as the more common sex, ranging from 50 to 072.7% [8,10,11,12,14,15,16,18,20,22], and two studies did not specify sex [17,19]. The mean ISS was 22, where reported. The most common mechanism of injury, where reported, was blunt in six studies [8,11,14,16,18,22] and burns in one study [15]. The most common methods to quantify the level of circulating NETs included nucleosome-calibrated (H3NUC) and free-histone-standardised (H3Free) ELISAs [8,11,12,13,14] and PicoGreen dsDNA assays [10,13,14,19] measured in plasma and/or isolated neutrophils. Table 1 outlines the key characteristics of the studies included.

### 4.1. Measurements from Plasma

Ten papers reported on NETs in patients’ plasma. NET components rather than whole NETs were measured in these studies. The NET components assessed included cfDNA, H3Cit and H3NUC, the concentrations of which are summarised in Table 1. Five papers assessed both cfDNA and H3Cit [8,12,13,14,15], one paper reported on H3Cit and H3NUC [11], and one paper measured all proteins present in plasma and extracted signatures of proteins in NETosis, rather than components of NETs themselves [21].

Seven papers included cfDNA as a proxy of NETs [8,10,12,13,14,15,18]. This was measured using ELISA in two papers [12,14], Pico-green fluorescence in four papers [8,10,15,18], and both in one paper [13]. Three papers used PCR [15,16,22]. While only four studies correlated cfDNA with outcomes [8,10,12,15], all seven papers compared trauma plasma cfDNA concentrations with healthy controls. In all studies, trauma patient plasma contained significantly more cfDNA than that in healthy controls; however, this was never more than two-fold higher than that in the healthy control concentration. 

Absolute values for cfDNA concentrations in trauma patients were only reported in four studies, ranging from 166–800 ng/mL, compared to those in the healthy control range,135–439 ng/mL [8,10,14,18]. In two papers, cfDNA was measured over time, demonstrating the fluctuation of concentrations across time periods varying from 10 days to 12 months, with concentrations in both studies peaking immediately after injury and at day 7 [10,15]. In two studies, cfDNA was measured when treated with DNase, which reduced cfDNA concentrations in a dose-dependent manner [14,18].

Six papers assessed H3Cit [8,11,12,13,14,15]. Five papers measured H3Cit via ELISA [8,11,12,13,14] and one via Western blot [15]. One paper quantified NETs via measuring DNA with Pico-green fluorescence in complex with MPO or H3Cit using ELISA [13].

Another paper quantified markers of NETosis or the NETosis pathway via the mass spectrometry of patient plasma [21].

### 4.2. Ex Vivo NETs

NETs were measured from neutrophils isolated from whole blood in four studies [13,16,17,20]. One paper measured basal NET production and observed enhanced generation at one hour post-injury compared to that in healthy controls. There was a significant reduction in NET production at 4–12 and 48–72 h post-injury [20].

Two papers examined NETosis capacity through the stimulation of neutrophils with phorbol myristate acetate (PMA) [16,20] or mitochondrial DNA (mtDNA) [17]. Two papers visualised NETs via their DNA using SYTOX green and fluorescence microscopy [16,20], as well as total cell-free DNA measurements in cell media with either qPCR [16] or total SYTOX green fluorescence [20]. One study visualised NETs with fluorescence microscopy using antibodies against neutrophil elastase and histone one (H1) [17].

One paper isolated neutrophils from TBI patients and co-labelled them with H3Cit and MPO to be quantified via flow cytometry or microscopy [13].

### 4.3. Correlation with Outcomes

MOF prediction was attempted in two studies [10,12]. One study found no evident correlation and the other study found that cell-free DNA (cfDNA) correlated with MOF scores only at day 2 and 6 post-injury (*p* < 0.05) [12]. Two studies aimed to predict post-injury sepsis [10,15], where cfDNA was elevated in those who developed sepsis between 5 and 14 days post-injury [15,18], and when combined with other additional inflammatory markers had prognostic potential to predict sepsis with the best discriminatory power at day 1 with an AUC of 0.921 [15]. One study aimed to predict and diagnose post-injury deep vein thrombosis (DVT) with NET markers found to be elevated in post-DVT plasma compared to those in pre-DVT and non-DVT plasma [8]. One paper included mortality as an outcome, finding that admission NETosis markers predicted mortality, longer ventilation requirements and ICU length of stay, with AUC values in the range of 0.69–0.89 [21]. Two papers evaluated the therapeutic relevance of treating NETs, however only in an ex vivo setting, finding that hyperbaric oxygen did not affect NETosis in trauma patient-derived neutrophils and that DNAse reduced NET markers in trauma patient plasma ex vivo [18]. One paper evaluated the pro-coagulant effect of NETs in traumatic brain injury (TBI) [15], finding that coagulopathic patients had increased NET markers compared to those in non-coagulopathic patients.

Of the papers that took measurements from plasma, three out of six papers demonstrated elevated H3Cit (3.5–6-fold) in trauma patients, ranging from 4.54 to 89.8 ng/mL, when compared to that in healthy controls, which ranged from 0.73 to 18.1 ng/mL, where reported [9,13,16]. One paper demonstrated elevated H3Cit in sepsis compared to that in trauma (*p* < 0.01) [6]. TBI patients had significantly more NETs than did healthy controls, and TBI patients with coagulopathy had significantly higher NETS than did those without coagulopathy [13].

Three of the ex vivo studies detected NETs in trauma patient samples and no NETs present in healthy control individuals’ samples (unless stimulated with PMA) [16,17,20].

One study used mass spectrometry to identify a number of proteins involved in NETosis rather than NET components themselves, which was predictive of poor outcomes [21].

## 5. Discussion

This systematic review evaluated the diagnostic, therapeutic and prognostic relevance of NETs in patients with severe trauma, with aims and findings illustrated in Figure 2. Few studies aimed to correlate NET markers with the study’s desired outcomes. A number of common themes were evident in the literature, including the relationship between plasma NET markers and coagulation, inflammation, endothelial dysfunction and the effect of DNAse. Overall, proxy markers, such as cell-free DNA, double-stranded DNA, citrullinated histone H3 and free histone H3, of NETs were measured at elevated levels in the trauma groups compared to those in the healthy controls. A wide variation in the methods used to measure and analyse NETs was identified without obvious recommendation or a consensus related to optimal techniques.

### 5.1. The Role of NETs in Post-Injury Hypercoagulation

It has been proposed that NETs act as a scaffold for clot formation in severe injury [23]. Liu et al., in 2021, measured elevated NET markers in patients with post-injury DVT; however, the predictive value of NETs is unclear from this study, as NET markers in those who did or did not s0ubsequently develop DVT were not compared [8]. Goswami et al., in 2022, found that thrombin formation in trauma plasma is accelerated in the presence of DNAse, suggesting that the degradation products of NETs may trigger coagulation, which poses some questions about the clinical utility of DNAse [14].

Jin et al. found that NETs were elevated in coagulopathic TBI patient plasma and that NETs were associated with levels of thrombin–antithrombin complexes, adding further support to the role of NETs in post-injury coagulation [13].

### 5.2. The Role of NETs in Inflammation in Trauma

Both McIlroy et al. and Itagaki et al. confirmed the presence of NETs after major trauma in comparison to that in controls. However, no quantitative assessment of NETs was used in either paper, making it difficult to discern the significance of their findings [16,17]. McIlroy et al. measured NETs via visually detecting mitochondrial DNA staining in neutrophils isolated from patient samples. While this paper provides a seminal description of NETs in major trauma, the employed methodology used may not have reliably differentiated between nDNA and mtDNA, as the MitoSOX red stain is specific to mtDNA when mtDNA is still within mitochondria. The study also concluded that there is a link between NETs and surgical timing through analysing peri-operative patient samples [22]. Itagaki et al. used microscopy images to document more NETs in trauma patients in comparison to those in controls. Younger patients with equivalent or less injury had higher levels of circulating mtDNA and more NETs as a baseline than those in elderly patients, as identified in microscopy images. Further investigation into the links between mtDNA and NETs may explain the mechanism behind the increased susceptibility to post-trauma organ failure in the elderly population [17]. Much of the literature views older age as having higher basal inflammation (“inflammageing”), so this result could suggest that either NETosis is an exception to the rule due to cellular senescence or an as-yet-unknown mechanism.

Schaid et al. identified NETosis to be elevated early on in severe injury and to be linked with post-injury mortality, a longer period of ventilator requirement and greater ICU requirement in trauma patients. The marker serpinB1 was described as a potential biomarker to identify excessive levels of NETosis post-injury, which may help in predicting the clinical trajectory of patients with severe trauma [21].

### 5.3. Therapeutics and Ex Vivo NET Formation

In the study conducted by Grimberg-Peters et al., hyperbaric oxygen treatment was found to potentially impair ROS-dependent pathways in neutrophils, and affect NET release. However, the reduction in NETosis capacity was only found in controls, not in trauma patients, which would suggest that either hyperbaric oxygen would not be therapeutic at reducing NETosis in trauma patients, or that the timing of sampling (<48 h of admission) may have confounded the findings [19].

Meng et al. demonstrated that DNAse degrades ex vivo-generated NETs in a concentration-dependent manner, and Goswami et al. demonstrated that NET markers in trauma patient plasma were reduced using DNAse, suggesting that DNAse degrades injury-induced NET structures [14,18]. However, since Goswami et al. also demonstrated increased thrombin generation after DNAse treatment, it is still unclear whether the degradation of NETs would be beneficial or if it would generate pro-inflammatory and pro-coagulant byproducts [14].

The main limitation of this review is the relative scarcity and diversity of publications that met the review criteria. The majority of available clinical studies used small sample sizes. In vitro or ex vivo samples analysed for the effects of interventions may also be insufficient in demonstrating the action and significance of NETs in the human microcirculation and interstitium. 

There are numerous methodological approaches to quantify NETs. Therefore, the variation among studies in terms of their quantification method can impact the analysis of their results, as well as the reliability of their findings [24]. The inconsistent methods of quantifying NETs in each study make it difficult to form a conclusion about the diagnostic, prognostic and therapeutic relevance of NETs in trauma patients. Only one of the studies reviewed quantitatively measured colocalised NET markers [13]. The varying centrifugation speeds for plasma generation used in each study also serve as an additional factor that could confound the ability to measure NETs, since intravascular NET structures are not well characterised as of yet, and the centrifugation speed that could remove NETs from plasma may include a wide range.

Citrullinated histones (H3Cits) are essential in the formation of NETs; thus, they are an important biomarker regarded as being highly specific for NETosis [24]. However, NETosis has also been shown to occur independently from the PAD4-dependent pathway depending on its stimulatory nature. Additionally, histones can be unstable in the blood, and H3 citrullination can occur in neutrophils which undergo autophagy, necrosis and apoptosis [25,26]. In the instance of trauma, mtDNA NETs have been demonstrated, and therefore mtDNA within NETs may be a more sensitive biomarker for quantification. It has been recently shown that the majority of post-injury cell-free mtDNA is found in larger structures, rather than free in the plasma, with NETs being a potential source [27].

Cell-free DNA (cf-DNA) has high objectivity and quantifiability when measuring NETs [25], but it can also be derived from other sources (i.e., necrosis and release from non-neutrophils, e.g., macrophages) [24]. PCR and dsDNA assays or alternative methods were also used to quantify the presence of NETs in some of the studies reviewed. The colocalisation of extracellular DNA and neutrophil-derived proteins is a method which offers the advantage of increasing the specificity of NETs [25]. This is demonstrated by the study conducted by Jin et al., who quantified NETs by detecting DNA complexes with multiple NET markers, including cfDNA, H3Cit, myeloperoxidase-DNA (MPO-DNA) and neutrophil elastase DNA (NE-DNA) [13].

## 6. Conclusions

As illustrated in Figure 2, NET markers have been found to be elevated in trauma patients in comparison to those in healthy controls. While NETs are linked to post-injury hypercoagulability, additional research is required to gain a deeper understanding of their role [8,13,14]. DNAse emerges as a promising therapeutic avenue for regulating post-injury organ failure with NETs, but further investigation related to its safety is warranted [18]. Additionally, the timing of injury relative to NET levels could be a crucial determinant in guiding optimal therapeutic interventions or in optimising its use as a clinical tool [17,21]. The quantifying methods of NETs in each study are inconsistent. There is a lack of large prospective descriptive in vivo studies to provide a comprehensive understanding of the natural history of NETosis after severe injury. Future research should be directed towards generating a standardised detection of the visualisation method for the quantification of NETs in pre-defined study populations for standardised clinically relevant outcome measures. Focused research can determine the actual role of this potentially crucial mechanism in post-injury inflammation and coagulation and their relevance to outcomes such as thrombo-embolic complications, multiple organ failure, sepsis and mortality. It can also establish the possible modalities of treatment that target a reduction in neutrophil-associated complications in polytrauma.

The presence of intravascular NETs has been demonstrated via the analysis of trauma patient plasma for NET components (DNA and NETs or NETosis-associated proteins). NETosis capacity has been measured in unstimulated or PMA-stimulated neutrophils isolated from trauma patient blood. NET markers have been consistently shown as elevated in trauma patient plasma compared to those in healthy controls, and correlations with coagulation, inflammation, endothelial dysfunction and outcomes of MOF, sepsis and mortality have been found. DNAse and hyper0baric oxygen have been studied as NET-targeted interventions; however, the benefits in trauma patients are still unclear. The heterogeneity of study populations, laboratory methods, data analysis and chosen outcomes indicate that further research and standardisation are needed to understand the clinical utility of NETs in trauma patients. 

## Figures and Tables

**Figure 1 biomolecules-13-01625-f001:**
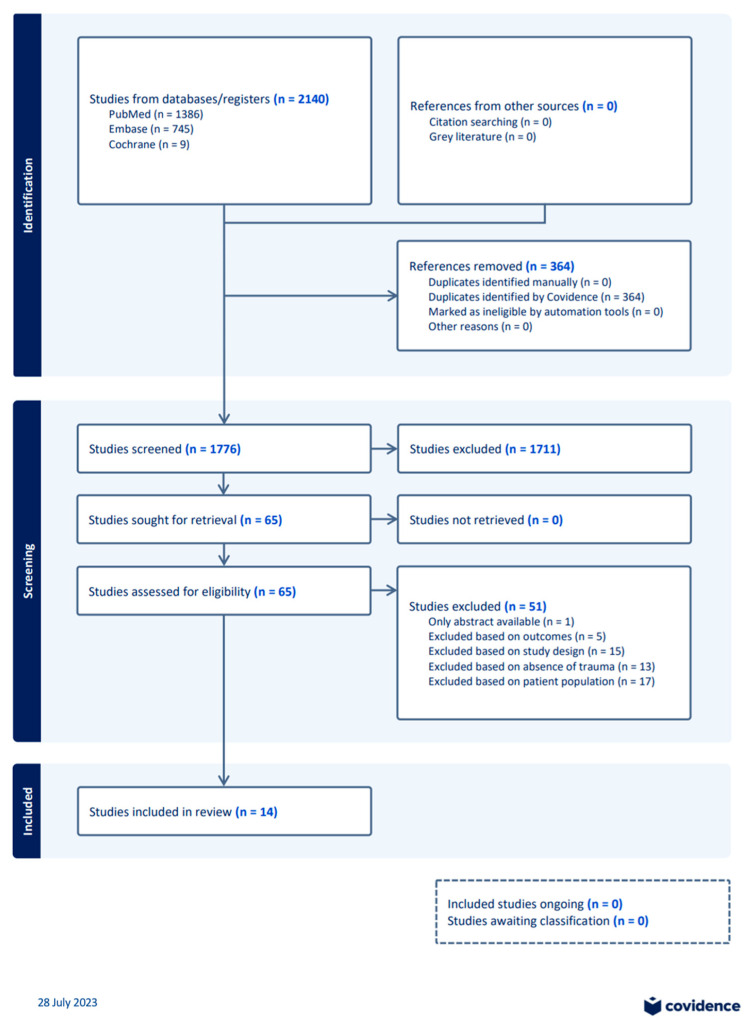
Selection process of the included publications.

**Figure 2 biomolecules-13-01625-f002:**
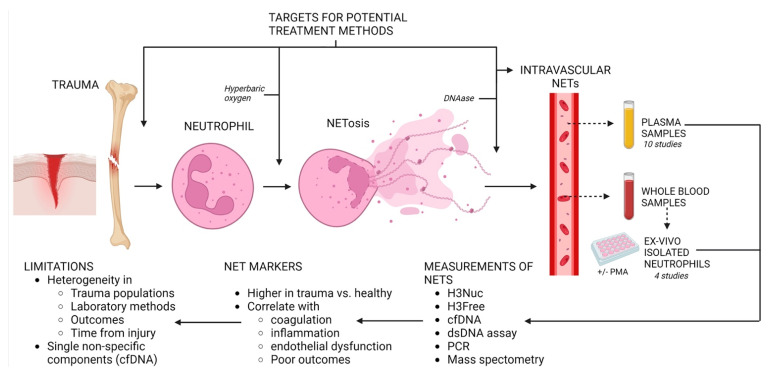
Post-Injury Neutrophil Extracellular Traps (NETs). The presence of intravascular NETs have been demonstrated via the analysis of trauma patient plasma for NET components (DNA and NETs or NETosis-associated proteins). NETosis capacity has been measured in unstimulated or PMA-stimulated neutrophils isolated from trauma patient blood. NET markers have been consistently shown as elevated in trauma patient plasma compared to healthy controls and correlations with coagulation, inflammation, endothelial dysfunction and outcomes of MOF, sepsis and mortality have been found. DNAse and hyperbaric oxygen have been studied as NET-targetted interventions, however the benefits in trauma patients are still unclear. The heterogeneity of study populations, laboratory methods, data analysis and chosen outcomes indicate that further research and standardisation are needed to understand the clinical utility of NETs in trauma patients. Figure created with Biorender.com.

**Table 1 biomolecules-13-01625-t001:** Summary of included papers.

Paper	Study Design	Sex (%male)	ISS	Quantification Method	cfDNA (ng/mL)	Histone H3 Proteins (ng/mL)
Goswami et al. 2022 [14]	Prospective cohort	67	12	H3Cit ELISAs, dsDNA assay	576.9	H3cit 4.54
Goswami et al. 2021 [11]	Prospective cohort	60	23	H3NUC, H3Free ELISAs.	N/A	H3NUC 89.8 H3free 5.74
Liu et al. 2021 [8]	Prospective cohort	51	N/A	H3NUC, H3Free ELISAs.	165.70	H3Cit 0.38
Chornenki et al. 2019 [12]	Retrospective observational combined cohort	73	≥16	H3NUC, H3Free ELISAs.	N/A	N/A
Hampson et al. 2017 [15]	Prospective cohort	80	Burn size > 15% of TBSA	dsDNA assay, Fluorescence microscopy, PCR, cfDNA	N/A	N/A
Margraf et al. 2008 [10]	Prospective observational cohort	65	>16	dsDNA assay	Isolated long bone trauma—160 Polytrauma—800	N/A
McIlory et al. 2014 [16]	Prospective cohort	50	13 +/− 7	cfDNA PCR	N/A	N/A
Itagaki et al. 2015 [17]	Clinical trial	Not specified.	23.8	Fluorescence microscopy	N/A	N/A
Hazeldine et al. 2019 [20]	Ongoing prospective longitudinal observational study	56	26	H3NUC, H3Free	N/A	N/A
Meng et al. 2012 [18]	Prospective cohort	72	38.8 +/− 2.6	dsDNA assay	334	N/A
Grimberg-Peters et al. 2016 [19]	Prospective Observational cohort study	Not specified	>16	dsDNA assay	N/A	N/A
McIlroy et al. 2018 [22]	Prospective cohort	81	18	Quantitative PCR	N/A	N/A
Schaid et al. 2022 [21]	Post-hoc analysis	80	>25	Mass spec	N/A	N/A
Jin et al. 2023 [13]	Prospective cohort	50	>16	ELISA dsDNA assay	N/A	N/A

H3NUC = nucleosome-calibrated histones; H3free = free histones; H3Cit = citrullinated histone 3 protein; cfDNA = cell-free DNA; PCR = polymerase chain reaction; TBSA = total body surface area; ISS = injury severity score. In studies where there are multiple time points, data are taken from the first time point documented. ISS is reported as the mean unless noted otherwise. NB: % male values are from the trauma cohort unless studies had mixed populations. dsDNA assays used DNA stains, PicoGreen or SYTOX.

## Data Availability

Data is not shared, reproducible strategy provided in the methods.

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
