# Peer review of "The Diagnostic, Therapeutic and Prognostic Relevance of Neutrophil Extracellular Traps in Polytrauma"

_biomolecules, 2023, doi:10.3390/biom13111625_

Round 1
Reviewer 1 Report
Comments and Suggestions for Authors
In the present systematic review Rogers et al., attempt to summarize literature related to neutrophil extracellular traps (NETs) formations and severe trauma patients. This is very relevant considering that current studies have now acknowledged the importance of NETs in not only in infection but also non-infectious diseases including cancers. However, the present review contains a lot of confusing sentences and is less succinct making it difficult to understand. Moreover, it also lacks coherence in some instances. Authors need to reorganize and format this review with tables where necessary so that it is accessible to readers. Here are few suggestions for the authors to take into consideration.
1) The primary and secondary outcomes are confusing. In one instance authors state that “Secondary outcomes included the relationship between NETs and post-trauma complications such as MOF” (line 72-73), in other instance author mention that “The primary outcome of interest was the relationship between NETs and a poor prognosis, mortality, and incidence of MOF. Secondary outcomes included potential interventions for NETs in trauma patients as well as the relationship between NETs and thrombus formation in traumatized patients” (line 106-109).
2) It would be useful to provide results in tabular form for the quality assessment for the studies included and excluded, criteria with scoring.
3) There are multiple instances where reference was entirely missing. Authors should ensure wherever a sentence related to the study is mentioned it needs to be cited and referenced.
Comments on the Quality of English LanguageEnglish is overall sound.
Author Response
Dear Reviewer,
Thank you for your insightful comments. Please find attached the detailed responses.

Reviewer 2 Report
Comments and Suggestions for Authors
The topic of the systematic review by Rogers at al. is for sure interesting for readers but the whole presentation of the topic is chaotic and very hard to understand. This already starts with the formal layout of the review with different font sizes used in different paragraphs (e.g. line 78-81, 120-133, 210-352). Additionally, the following issues need to be addressed:
- SMART used for Figure 2 should be cited
- Supplement not really necessary, could also be included in the main text
- study selection too many spaces after period? Also, some other parts seem to have too much space in between words (e.g. l. 166).
- Abstract: 8 plasma samples, 3 whole blood samples; results: ten paper measured NETs from plasma, 4 ex vivo, which one is true?
- table 1 is a bit confusing:
1) H3cit (ng/ml) is given as table header but in the column, it sometimes says H3NUC and sometimes H3Cit in the column. Maybe use a less specific term in the header? Quantification method: is H3free same as H3cit?
2) sometimes H3cit is used and other times citH3 (e.g. Hampson et al. 2017 (17)), please be consistent
3) Also Fixed PMN is not really a quantification method, maybe it would be useful to have another column to the table representing the type of sample used?
4) It is also not clear why patients were sometimes further divided in the column Sex (%male)
5) the relationship between sepsis and traumatic injury should be explained in the introduction.
6) Introduction in general is very limited on trauma, more focused on neutrophils. Readers new to trauma research will not understand.
7) Liu et al. 2021(9) ISS emailed author, waiting reply should be changed to n.a.
8) top of page 6, where does total trauma = 71.8% M. and sepsis 334 belong to? Meng et al. 2012 (20) from bottom of page 5?
9) Abbreviations below table 1 not all used in table, e.g. RCT, TOI, MODS, HBO, PMA, RFU, THR, PNM (typo?), HC;
10) abbreviations used but not explained: ISS, cfDNA, H3cit/citH3, dsDNA
11) Is dsDNA assay always quant-iT PicoGreen or only for Jin et al. 2023 (15)?
12) Why are so many values N/A if e.g. cfDNA or H3NUC, H3free was measured as indicated in the table?
- L. 154 what is DVT? Please explain abbreviation
- Text is written free histone standardized (H3Free) ELISAs (l. 142) but table and later in text it is citH3 (e.g. l. 163), difference between H3NUC and citH3, H3free should be explained
- L. 189: Pico-green fluorescence?
- L. 193/194 comma missing?
- L.197/198 sentence doesn’t make sense
- L. 202 NETS
- L. 202/203 what means detection by neutrophil elastase and H1 antibodies? Also IF?
- L. 201 abbreviation mtDNA?
- L. 207 of?
- Discussion: L. 213/214 they were also evaluated ex vivo for the effectiveness of therapeutics on NET formation, where?
- L. 220 onwards: Hypercoagulability was associated with NETs constituents, nowhere mentioned in results part
- L.279-282: sentence to complicated and difficult to understand
- L.282 how does the use of patient plasma make this study incomparable if most of the studies used plasma instead of whole blood?
- L.284 centrifugation of samples to obtain plasma is very common practice, how did the other studies obtain plasma?
- L. 286/287 what are the higher scoring markers? What does this mean?
- L.310 is this not true for all PicoGreen based measurements?
- Figure 2 is not really a conclusion or understandable as the Figure legend does not offer any explanation of the figure, also the authors pointed out that some studies used ex-vivo NET formation in their studies, which is missing from the figure as only intravascular NETs are used for the measurements according to their figure
Comments on the Quality of English LanguageIt seems like the text was written by completely different people and not checked afterwards. Some sentences do not make sense and are very poor English, whereas others are very well written. The whole text should be checked carefully for any typos, missing words, commas, grammar etc.
Author Response
Dear Reviewer,
Thank you for your thoughtful comments. Please see our detailed responses attached.

Round 2
Reviewer 1 Report
Comments and Suggestions for Authors
The authors have addressed all my comments. Manuscript can be considered for publication.
Comments on the Quality of English LanguageNo major editing is required.
Author Response
Dear Editor and Reviewers.
Please see the responses attached.
Reviewer 2 Report
Comments and Suggestions for Authors
The new version of the review is significantly improved. There are only a few minor things that still need to be addressed.
- citation of SMART was not found in text
- Figure Legend/Description for Figure 2 is still missing
- Abstract (l. 20) and Results (l. 138) mention 11 samples from plasma and 3 samples from whole blood but Results-measurement from plasma (l. 156) and Figure 2 state 10 samples from plasma and 4 from whole blood. Which one is correct?
- L. 177/178 citation is missing
- If understood correctly, the new version does not contain supplementary data, so the sentence in lines 338/339 should be removed.
Comments on the Quality of English LanguageEnglish language is fine.
Author Response
Daer Editor and Reviewer please see our detailed responses and revisions of the manuscript attached.
